# Effect of Plasma-Enhanced Atomic Layer Deposition on Oxygen Overabundance and Its Influence on the Morphological, Optical, Structural, and Mechanical Properties of Al-Doped TiO_2_ Coating

**DOI:** 10.3390/mi12060588

**Published:** 2021-05-21

**Authors:** William Chiappim, Giorgio Testoni, Felipe Miranda, Mariana Fraga, Humber Furlan, David Ardiles Saravia, Argemiro da Silva Sobrinho, Gilberto Petraconi, Homero Maciel, Rodrigo Pessoa

**Affiliations:** 1Laboratório de Plasmas e Processos, Instituto Tecnológico de Aeronáutica, Praça Marechal Eduardo Gomes 50, São José dos Campos 12228-900, Brazil; giorgiotestoni@gmail.com (G.T.); mirannda.fs@gmail.com (F.M.); argemiro@ita.br (A.d.S.S.); petra@ita.br (G.P.); odairtur@gmail.com (H.M.); 2i3N, Departamento de Física, Universidade de Aveiro, Campus Universitário de Santiago, 3810-193 Aveiro, Portugal; 3Instituto de Ciência e Tecnologia, Universidade Federal de São Paulo, Rua Talim 330, São José dos Campos 12231-280, Brazil; 4Centro Estadual de Educação Tecnológica Paula Souza, Programa de Pós-Graduação em Gestão e Tecnologia em Sistemas Produtivos, São Paulo 01124-010, Brazil; humber@fatecsp.br; 5Laboratoire TIMA, Université Grenoble Alpes, 38000 Grenoble, France; davidcesar@yahoo.com; 6Instituto Científico e Tecnológico, Universidade Brasil, São Paulo 08230-030, Brazil

**Keywords:** plasma-enhanced atomic layer deposition, titanium dioxide, aluminum oxide, nanolaminates, superstoichiometry, doping

## Abstract

The chemical, structural, morphological, and optical properties of Al-doped TiO_2_ thin films, called TiO_2_/Al_2_O_3_ nanolaminates, grown by plasma-enhanced atomic layer deposition (PEALD) on p-type Si <100> and commercial SLG glass were discussed. High-quality PEALD TiO_2_/Al_2_O_3_ nanolaminates were produced in the amorphous and crystalline phases. All crystalline nanolaminates have an overabundance of oxygen, while amorphous ones lack oxygen. The superabundance of oxygen on the crystalline film surface was illustrated by a schematic representation that described this phenomenon observed for PEALD TiO_2_/Al_2_O_3_ nanolaminates. The transition from crystalline to amorphous phase increased the surface hardness and the optical gap and decreased the refractive index. Therefore, the doping effect of TiO_2_ by the insertion of Al_2_O_3_ monolayers showed that it is possible to adjust different parameters of the thin-film material and to control, for example, the mobility of the hole-electron pair in the metal-insulator-devices semiconductors, corrosion protection, and optical properties, which are crucial for application in a wide range of technological areas, such as those used to manufacture fluorescence biosensors, photodetectors, and solar cells, among other devices.

## 1. Introduction

Artificial intelligence, robotics, the cloud, and the internet of things, terms that were unknown a few years ago, are now part of everyday life for all of us. These are technologies that are part of an established concept in the industrial sector: Industry 4.0, also called the 4th Industrial Revolution. This phenomenon is changing, on a large scale, automation and data exchange, production steps, and business models through machines and computers. Innovation, efficiency, and customization are the keywords to define the concept of Industry 4.0. However, to advance, the evolution of a whole chain of technologies is necessary. For example, it is essential to improve nanochips that integrate the entire computer automation system that controls and manages this industry. It is required to enhance the bio and nanosensors to control industries, businesses, and people. It is also necessary to improve the anti-corrosion coatings used to protect all devices that are part of this novel industry.

In this sense, the development of devices applied to industrial processes [1], home security [2], monitoring of air quality [3] and explosives [4], and the detection of pollutants [5] and toxic compounds [6] are essential, in addition to wearable devices used as accessories and implants [7]. The manufacturing process of devices can involve a broad range of deposition techniques, such as sol-gel process [8,9,10,11,12], sputtering [13,14], chemical vapor deposition (CVD) [15], plasma spray [16,17], microwave-assisted synthesis [18,19], and the disruptive technique atomic layer deposition (ALD) [20,21,22]. Among them, ALD stands out for its unique capabilities, which include the complex shapes coverage embedded in high conformal 3D areas [23], the growth of stacked monolayers of different nanomaterials [24], and the growth of thin films precisely defined by self-limited surface reactions [25,26,27,28,29]. The ALD’s versatility allows its application in a broad range of fields, such as micro and nanoelectronics [30,31], biomedical engineering [32], on food packaging against corrosion [33,34], fuel cells [35], solar cells [36], anti-tarnish coatings on jewels surfaces [37], smart textiles [38], membranes, and optoelectronics [39,40,41]. Despite the wide range of applications of the ALD technique, fundamental studies are needed to understand essential aspects of the chemical, morphological, mechanical, and optical properties of the thin films that influence devices’ properties and their applications.

For example, roughness, which is a morphological property, can be used to control the mobility of pair hole-electron in metal-insulator-semiconductor (MIS) devices [42], which is essential control to enhance the power-conversion efficiency (PCE) on MIS solar cells devices [43]. A mechanical property considered fundamental in all devices and sensors is the protection against corrosion, which increases the lifetime of the devices [44]. On the other hand, sensors based on optical parameters can be fabricated through control of refractive index, which is an optical property that can be used to produce a high-quality resonant waveguide grating (RWG) that is used to fabricate fluorescence biosensors, and photodetectors, beyond other devices [45]. Therefore, it is essential to understand the fundamental properties of thin films.

In this contribution, the impact of the insertion of the Al_2_O_3_ monolayers onto the TiO_2_ thin films grown by PEALD (in order to form a TiO_2_/Al_2_O_3_ nanolaminate structure) was studied in detail. TiO_2_ and Al_2_O_3_ were chosen since they have been the most widely studied ALD materials [46,47,48,49,50,51,52]. Moreover, the PEALD technique was selected because it is widely used for deposition layers in a wide range of sensing devices [53,54,55]. It is worth mentioning that ALD TiO_2_ [56,57], ALD Al_2_O_3_ [58,59], as well as ALD TiO_2_ doped by Al [60,61,62] have been extensively studied in the past. However, novel contributions appear with each new modification added to the structure of the films, which justifies the present study. Therefore, to infer the effects of the PEALD nanolaminates, several structural, chemical, morphological, mechanical, and optical characterization techniques were used, namely, Rutherford backscattering spectroscopy (RBS), Raman spectroscopy, atomic force microscopy (AFM), nanoindentation, and spectroscopic ellipsometry.

It is worth mentioning that all the present work results are compared with our previous work [24], where thermal ALD was used (that is, the H_2_O vapor was used as an oxidizer). Nevertheless, due to the difficult to compare different structures grown by a broad range of teams, reactors, types of stacks of TiO_2_/Al_2_O_3_, and parameters. In Section 3.4, we compare our results with previous studies by other research teams in order to better discuss the differences between the Al-doped TiO_2_ films reported in the literature.

## 2. Materials and Methods

### 2.1. Synthesis of Al-Doped TiO_2_ (TiO_2_/Al_2_O_3_ Nanolaminates)

A TFS-200 ALD system from Beneq (Beneq Oy, Espoo, Finland) used in plasma mode was employed to deposit TiO_2_/Al_2_O_3_ nanolaminates films. The schematic representation of the plasma enhanced ALD (PEALD) reactor is presented in Figure 1.

The system was operated with a capacitively coupled plasma (CCP) reactor at a pressure lower than 1.0 Pa and gas pressure of 1.0 hPa through the insertion of 300 sccm of N_2_ [24,29] with the substrate holder grounded. The plasma was generated by capacitive coupling in the upper plate on the plasma generation zone with an RF power supply of 13.56 MHz (Cesar, Advanced Energy Inc., Fort Collins, CO, USA) with the bottom grid electrode and radial reactor walls grounded. TiO_2_/Al_2_O_3_ nanolaminate films were grown on three-inch p-type <100> Si wafers (University Wafer Inc., South Boston, MA, USA) and commercial SLG glass (Sigma-Aldrich, São Paulo, Brazil) at a process temperature of 250 °C. All substrates were cleaned into an ultrasonic bath with a solution of deionized water and acetone (99.55%, Sigma-Aldrich, São Paulo, Brazil) for 5 min and, subsequently, dried through nitrogen (N_2_) gas (99.9%, White Martins, Jacareí, Brazil). The design of the TiO_2_/Al_2_O_3_ nanolaminates followed the recipe of alternating cycle(s) of Al_2_O_3_ and TiO_2_ in supercycle, according to [24]. It worth mentioning that O_2_ gas (99.99%, White Martins, Jacareí, Brazil) was used to generate the oxidant precursor (O_2_ plasma), differently from our previous work [24] that used H_2_O vapor and grown the films on a low-volume crossflow type reactor operating in thermal mode (Figure 2). Both works used titanium tetra-isopropoxide (TTIP) (≥97%, Sigma-Aldrich, São Paulo, Brazil) at 70 °C and trimethylaluminum (TMA) (97%, Sigma-Aldrich, São Paulo, Brazil) as metallic precursors of TiO_2_ and Al_2_O_3_, respectively.

Table 1 summarizes the supercycle and the corresponding pulse ratio utilized in this work. TiO_2_/Al_2_O_3_ nanolaminate films were grown under the following conditions of pulse ratio ([Al]/[Al + Ti]): 0 (sample 0% Al(P) (TiO_2_)); 0.004 (sample 0.4% Al(P)); 0.012 (sample 1.2% Al(P)); 0.016 (sample 1.6% Al(P)); 0.032 (sample 3.2% Al(P)); and 1 (Al_2_O_3_ (P) sample). From our previous work using thermal ALD [24], the samples that were grown using the same pulse ratio condition were labeled as 0% Al(T); 0.4% Al(T); 1.2% Al(T); sample 1.6% Al(T); 3.2% Al(T); and Al_2_O_3_ (T), respectively. The (P) represents samples grown on plasma mode, and (T) represents samples grown on thermal mode.

### 2.2. TiO_2_/Al_2_O_3_ Nanolaminate Films Characterizations

Rutherford backscattering spectroscopy (RBS) was used to check the nanolaminate film thickness (indirectly) and elemental film composition (in at. %) (directly) on p-type <100> Si. The experiment was carried out in a Pelletron accelerator (2.2 MeV 4He + beam) with the particle detector positioned at an angle of 170° to the incident beam. The detection sensibility of the measurement to Ti, O, Al, and Si is about 5% [30]. The thickness of the films was calculated indirectly with the help of MultiSIMNRA software [63], converting the RBS density values (10^15^ atoms.cm^−2^) into the thickness (nm) of the layer. The indentation hardness, indentation modulus, and Young’s modulus are fundamental mechanical properties of the TiO_2_/Al_2_O_3_ nanolaminates grown on p-type <100> Si that was measured by microhardness tester FM-700 (Future Tech, Kawasaki, Japan). Nanoindentation was carried out at a depth of up to 30% of the film thickness, across the area of around 25 µm^2^. All samples were characterized eight-fold in a 2D array at different points. Load and unload profiles were evaluated through the Oliver–Pharr method [64,65]. Raman scattering was used to identify the Raman-active modes for the TiO_2_/Al_2_O_3_ nanolaminates crystallinity on p-type <100> Si. The Raman spectra were acquired at 25 °C, using a Raman micro spectrometer model Horiba-evolution (Horiba, Kyoto, Japan) supplied with a multichannel charge-coupled device detector thermoelectrically cooled. The excitation wavelength was 532 nm with the incident laser beam power <10 mW. Between the range from 100 to 900 cm^−1^, the spectral resolution was better than 1 cm^−1^, and the Voigt profile was used to analyze the phonons modes by fitting Raman peaks. An atomic force microscope (AFM) model SPM9500 J3 (Shimadzu, Tokyo, Japan) was used for morphological characterization of TiO_2_/Al_2_O_3_ nanolaminates on p-type <100> Si. AFM characterization was performed in air using Si cantilevers with a tip radius ≤ 10 nm in a surface area of 1 × 1 µm^2^ and 5 × 5 µm^2^, and the images were treated by Gwyddion data analysis software [66]. Optical transmittance T(λ) and reflectance R(λ) were performed through UV–Visible–NIR spectrophotometer model V-570 (Jasco, Easton, MD, USA) equipped with an integrating sphere. T(λ) and R(λ) spectra of TiO_2_/Al_2_O_3_ nanolaminates on commercial SLG glass were measured over the wavelength range from 220 to 2000 nm. Spectroscopy ellipsometry was performed with the model Uvisel Jobin-Yvon (Horiba, Kyoto, Japan). The data set acquired from ellipsometry was used to calculate the bandgap of the TiO_2_/Al_2_O_3_ nanolaminates grown on commercial SLG glass, and these data were treated by the Tauc plot method [25].

## 3. Results and Discussion

### 3.1. Chemical Composition, Thickness, and Growth per Cycle

RBS analyses were used to study the elemental chemical composition (ECC), thickness, and growth per cycle (GPC (nm/cycle)) of TiO_2_/Al_2_O_3_ nanolaminates grown on p-type <100> Si (as tabulated in Table 2).

As shown in [25,29], the ECC, thickness, and GPC are approximately equal on both substrates, p-type <100> Si and SLG. Therefore, the RBS was performed only on p-type <100> without affecting the research. 0.4% Al(P) and 1.2% Al(P) samples show a small percentage of Al_2_O_3_. Thus, the TiO_x_/Al_2_O_3_ films present an overabundance of oxygen on the surface, i.e., x values of 2.28 ± 0.01 and 2.06 ± 0.01, respectively, for 0.4% Al(P) and 1.2% Al(P) samples. These results differ from those described for thermal ALD in our previous work [24] and are presented in Table 1. For the same deposition parameters, i.e., 0.4% Al(T) and 1.2% Al(T) samples were obtained x values of 1.50 ± 0.01 and 1.78 ± 0.01, respectively. On the other hand, when O_2_ plasma is used as an oxidant source, the superstoichiometry appears. Recently, Wei et al. [67], Raztsch et al. [68], Bousoulas et al. [69] and Chiappim et al. [27,29,30] using different parameters such as temperature, pressure, power source, reactors, ALD pulse time, and substrates showed an excess of oxygen in their TiO_x_ films.

Figure 3 presents a schematic representation that describes the mechanism responsible for an overabundance of oxygen on the surface of the films. Firstly, the oxygen ions from O_2_ plasma imping distinct depth into the film [70] and through the physical diffusion mechanism [71,72] the oxygen reach the surface of the film being the oxygen consumed partially by Ti precursor. Finally, a new ALD cycle starts with the oxygen stored on the subsurface of the nanolaminates. As can be seen, the oxygen in excess can be partially removed when exposed to the Ti precursor. This observation suggests that the superstoichiometry oxygen present in TiO_2_ film is in a reactive state that corroborates with the capability of the oxygen being consumed by reaction with the Ti precursor. On the other hand, due to the short duration of the ALD cycle, a portion of the oxygen is confined to the film in each period. This mechanism creates continuous superstoichiometry in each cycle, generating a higher saturated growth per cycle (GPC), and, consequently, a higher thickness than the nanolaminates grown in thermal mode (as shown in Table 2).

According to Schneider et al. [73], the film surface acting as a reservoir of oxygen corroborates with the rise of the growth of more than a monolayer in each cycle. They postulated that a diffuse physical mechanism of oxygen uptake and consumption into and out of the film is responsible for increases the growth due to vacancy oxygen and the reaction of diffusion that can advance continuously even after a monolayer is deposited. On the other hand, increasing the amount of Al_2_O_3_ in the films reduced the x values of 1.89 ± 0.01 and 1.50 ± 0.01 in 1.6% Al(P) and 3.2% Al(P) samples, respectively. This suggests a total oxygen consumption for a higher doping of TiO_2_ with Al_2_O_3_. Therefore, the PEALD modified the nanolaminates’ structure, and this behavior is evident when compared to nanolaminates grown by thermal ALD (as showed in Table 2). It worth highlighting that all characterizations of the nanolaminates were performed in different positions of the samples showing that the oxygen excess is not localized.

### 3.2. Structural, Morphological, and Mechanical Properties

Figure 4 shown the Raman spectra of TiO_2_/Al_2_O_3_ nanolaminate films grown on p-type <100> Si, and as a benchmark, it was added a Raman spectrum of 0% Al(P). On 0% Al(P), 0.4% Al(P), and 1.2% Al(P) were observed four Raman-active modes associated with anatase structure, namely, A_1g_ (519 cm^−1^), B_1g_ (397 cm^−1^), and E_g_ (144 and 636 cm^−1^). All Raman spectra presented a strong peak at 144 cm^−1^ [74]. These Raman-active modes evidenced a crystalline structure in the anatase phase [25] for 0% Al(P), 0.4% Al(P), and 1.2% Al(P) samples showing that a lower amount of Al_2_O_3_ doping the TiO_2_ films preserved the characteristics of TiO_2_ films. On the other hand, a slight increase in the amount of Al_2_O_3_ changed the crystalline phase to amorphous, i.e., an increase of pulse ratio from 0.012 to 0.016 was sufficient to shift the crystallinity drastically.

Table 3 summarizes the full width at half maximum (FWHM), the integrated area, and Raman peak position of 0% Al(P), 0.4% Al(P), and 1.2% Al(P) samples. The full width at half maximum (FWHM) no suffers change. Therefore, according to Bassi et al. [75], the crystal size of anatase TiO_2_ stays approximately constant. The integrated area decreases with the increase of Al_2_O*_3_* amount and disappears at a pulse ratio of 0.012. Peak position presents a blue shift (Raman peak position shift to the higher wavelength side) in 0% Al(P), 0.4% Al(P), and 1.2% Al(P) samples for all Raman-active modes. Parker et al. [76] related the bandwidth shift to non-stoichiometry, and Ratzsch et al. [68] showed the shift of position peak for the E_g_ peak at 144 cm^−1^ and associated a slight excess in oxygen (O/Ti > 2) in high-density TiO_2_ with large crystallites embedded in the amorphous matrix layer. Therefore, the Raman-active modes blueshift observed in Figure 4 for 0% Al(P), 0.4% Al(P), and 1.2% Al(P) samples can be attributed to the overabundance of oxygen on the surface of the films, corroborating with RBS results (Table 2). It worth noting from our previous work [30] that a redshift in E_g_ (144 cm^−1^) Raman-active mode of TiO_2_ films (superstoichiometric) grown by PEALD was observed. This change in the present work can be related to the crystalline growth of the TiO_2_/Al_2_O_3_ nanolaminate proposed in [24]. This behavior indicates that the crystalline growth embedded into the amorphous phase can be controlled by the doping effect of TiO_2_ by insertion of Al_2_O_3_ monolayers, which may have interesting industrial applications due to the control from redshift to blueshift, through the change of Al_2_O_3_ concentration in TiO_2_/Al_2_O_3_ nanolaminates.

Figure 5 and Figure 6 present AFM images and RMS roughness of the TiO_2_/Al_2_O_3_ nanolaminate surface grown on p-type <100> Si, respectively. As can be seen, in the case of 0.4% Al(P) and 1.2% Al(P) samples (Figure 5a–d) various sizes of the crystallites can be observed [24]. Figure 5e–h shows the inhibition of crystallization using as doping of a monolayer of Al_2_O_3_ in every 60 monolayers of TiO_2_ ([Al]/[Al + Ti] = 0.016).

Hence, besides ALD process parameters, such as substrate temperature, oxygen gas flow rate, and plasma power [68,77], incorporating thin Al_2_O_3_ layers into the TiO_2_ film suppress the crystallinity of TiO_2_ films without decrease the quality of the films [78]. Figure 6 shows that with the decrease in crystallinity, the surface roughness of TiO_2_/Al_2_O_3_ nanolaminates undergoes a considerable reduction, which was observed in previous works carried out in thermal ALD [24]. Due to the superstoichiometry of 0.4% Al(P) and 1.2% Al(P) samples, the surface roughness is more significant than the surface roughness of 0.4% Al(T) and 1.2% Al(T) samples that were grown in thermal ALD [24]. Thus, the AFM characterization corroborates with the Raman spectra (Figure 4), showing that the doping effect in the proposed PEALD TiO_2_/Al_2_O_3_ nanolaminate can adjust the morphological parameters.

Figure 7 compares the surface hardness, the indentation modulus, and Young’s modulus of the TiO_2_/Al_2_O_3_ nanolaminates on p-type <100> Si measured in an indentation load of 0.1–0.4 mN. PEALD results were compared to thermal ALD [24]. Figure 7a shows that the 0% Al(P) sample (TiO_2_) presents softer mechanical properties (4.5 GPa). According to Mohammed et al. [79], this behavior occurs in anatase TiO_2_ films. On the other hand, the Al_2_O_3_ (P) sample presents more hardened mechanical properties (10.3 GPa), as shown by Tripp et al. [80] that found a hardness of ALD Al_2_O_3_ films of 12.3 GPa. As shown in Figure 7a, the surface hardness reaches the more hardened mechanical properties when increasing the amount of Al_2_O_3_ into the TiO_2_ films. Hence, we can suggest that when the TiO_2_/Al_2_O_3_ nanolaminate becomes amorphous, its mechanical properties become harder. This is in accordance with Coy et al. [81]. In all samples grown by PEALD, the surface hardness is higher than the films produced under the same conditions by thermal ALD [24].

The enhancement of the mechanical properties through controlled doping of TiO_2_ by insertion of Al_2_O_3_ monolayers improved the physical properties towards practical applications. From the technological perspective, the TiO_2_/Al_2_O_3_ nanolaminates should be resilient to damage due to impact or mechanical stress and wear. Therefore, the control and proper assessment of mechanical properties would define the future applicability and the potential implementation in Industry 4.0. Figure 7b,c show the same behavior of the indentation and Young’s modulus. In the case of thermal ALD [24], the values agree with the experimental values reported in the literature for anatase TiO_2_ thin films [82]. On the other hand, in the present work, the 0.4% Al(P) and 1.2% Al(P) samples showed values (152 and 310 GPa, respectively) above the anatase TiO_2_ values reported in the literature (151 GPa) [82].

Therefore, it can be suggested that the surface hardness (Figure 7a) is dependent on the increase of the amount of Al in the nanolaminates. On the other hand, it can be suggested that the indentation and Young’s modulus are dependent on the superstoichiometry, as can be seen in 0.4% Al(P) and 1.2% Al(P) samples (Figure 7b,c), where occurs a sudden increase.

### 3.3. Optical Properties

Figure 8 shows the optical properties carried out by UV–Vis spectrophotometry and spectroscopic ellipsometry of the TiO_2_/Al_2_O_3_ nanolaminates grown on SLG glass. As shown in Figure 8a, the average transmittance is 60–70% is more significant than the average of the TiO_2_/Al_2_O_3_ nanolaminates (50–60%) grown by thermal ALD [24]. Another observation is related to the shift in the maximum transmittance on a range of 350–450 nm wavelengths for thermal ALD, which it shifted to 460–750 nm wavelengths to PEALD. According to Sreemany and Sen [83], the increase of thickness can be responsible for a shift; however, the 1.2% Al(T) sample (thermal mode) with 83 nm of thickness showed an average transmittance near the 0.4% Al(P), 1.2% Al(P), 1.6% Al(P) and 3.2% Al(P) samples (~160 nm). Hence, we can suggest that this behavior occurs due to the stoichiometry of the 1.2% Al(T) ALD sample, which is equivalent to the PEALD samples with Al(%) of up to 0.6% in bulk. These behaviors mentioned above are fundamental for applications in optical sensors, as the controlled doping of PEALD TiO_2_ by the insertion of Al_2_O_3_ monolayer changes this optical parameter [84]. Figure 8b shows the reflectance spectra as a complementary study of transmittance spectra and shows a concordance between the results.

Figure 8c shows the optical losses, an essential parameter to design thin-film optical sensors that should have as small losses as possible. Optical losses are calculated from 100%-transmittance-reflectance (100%-T-R) [78]. Absorption and scattering reduce the intensity of transmitted light, causing these losses, and the spectrophotometry results showed the homogeneity of the nanolaminates and low optical losses (<10%). The refractive index of PEALD nanolaminates varied from 2.98 to 2.65 (plot not shown). The decrease of refractive index on TiO_2_/Al_2_O_3_ nanolaminates is due to the lower value of the refractive index of Al_2_O_3_ (1.62). Therefore, these results show that it is possible to adjust the refractive index by incorporating amorphous Al_2_O_3_ to TiO_2_ with low optical losses (similar results were showed by Ghazaryan et al. [78]).

The indirect optical band gap was calculated according to [24], and the results are shown in Figure 8d. A slight increase in the indirect band gap was observed when the amount of Al_2_O_3_ increased from 0.004 to 0.032 (pulse ratio). According to Scanlon et al. [85], the transition from anatase TiO_2_ to amorphous TiO_2_ increases the band gap. Comparing the thermal ALD [24] and PEALD results, it was observed that PEALD nanolaminates have an indirect band gap below 3.5 in all samples (0.4% Al(P), 1.2% Al(P), 1.6% Al(P), and 3.2% Al(P)) and in the case of thermal ALD nanolaminates it has an indirect band gap above 3.5. This behavior shows that in addition to controlling the optical parameters through the doping of TiO_2_ films, the PEALD process is crucial to change this optical parameter.

### 3.4. Results Comparison with Previous Studies by Other Research Teams

This section summarizes the main works related to fundamental studies with Al-doped TiO_2_ and stacks of TiO_2_/Al_2_O_3_ films, both called nanolaminates. The real challenge was to find works from other teams that could be compared to our current work. Our first challenge is related to the fact that each type of TiO_2_/Al_2_O_3_ stacking and all types of Al doping form unique films with specific properties, structures and growth forms. Another difficulty faced is due to the large number of articles that study the application of these materials (more than 100 works were found), ranging from microelectronics [54,61,62,84] to application in tunable color coating [38], which makes it difficult to compare with the present work. Finally, we found about 11 works that can be compared. These works vary between the years of 2004 and 2021 and are summarized in Table 4 [49,86,87,88,89,90,91,92,93,94]. Among these works, only one work that used O_2_ plasma can be compared with our characterizations [60], that is, there is a lack of works with TiO_2_/Al_2_O_3_ nanolaminates that study fundamental properties and that use O_2_ plasma as a ligand precursor. Therefore, further studies of PEALD are necessary for the growth of nanolaminates.

## 4. Conclusions

Oxygen plasma used as oxidant precursor, and the controlled doping of TiO_2_ by the insertion of Al_2_O_3_ monolayers induced chemical, structural, morphological, and optical modifications in the PEALD TiO_2_/Al_2_O_3_ nanolaminates. The morphological modification is essential to control the mobility of the pair hole-electron in metal-insulator-semiconductor devices, being crucial to enhance the power-conversion efficiency in MIS solar cells. The property called protection against corrosion is essential to increases the device’s lifetime. The optical property improvement can be used to produce a high-quality resonant waveguide grating used to fabricate fluorescence biosensors and photodetectors beyond other devices. The 0.4% Al(P) and 1.2% Al(P) samples show an overabundance of oxygen on the surface of the films, and a schematic representation was included that describes this mechanism based on O_2_ plasma action. A slight increase in the amount of Al_2_O_3_ caused a lack of oxygen in the 1.6% Al(P) and 3.2% Al(P) samples, which suggests a total consumption of oxygen for greater doping of TiO_2_ by Al_2_O_3_. Raman spectra and AFM images show that the samples with an overabundance of oxygen (0.4% Al(P) and 1.2% Al(P)) are crystalline with four Raman-active modes associated with anatase structure, namely, A_1g_ (519 cm^−1^), B_1g_ (397 cm^−1^), and E_g_ (144 and 636 cm^−1^). 1.6% Al(P) and 3.2% Al(P) samples with a lack of oxygen are amorphous. This shown that a slight increase in the amount of Al_2_O_3_ changed the crystalline phase to amorphous, i.e., an increase of pulse ratio from 0.012 to 0.016 was sufficient to shift the crystallinity drastically. The controlled doping of metal oxide thin films by ALD method could have interesting industrial applications due to the control from redshift to blueshift in Raman-active modes, for the case of this work, through the change of Al_2_O_3_ concentration in TiO_2_/Al_2_O_3_ nanolaminates. The transition from the crystalline to the amorphous phase increases the surface hardness of the PEALD TiO_2_/Al_2_O_3_ films and becomes more resilient to damage due to impact or mechanical stress and wear. The optical properties results showed that it is possible to tune the refractive index by incorporating amorphous Al_2_O_3_ into TiO_2_ with low optical losses with the transition from anatase to amorphous TiO_2_ and increasing the band gap. Therefore, this study is a promising proof-of-principle that the controlled doping of TiO_2_ by Al_2_O_3_ could be used to pave the way for the fabrication of specific wavelength optoelectronic devices operating in the visible range, alongside other devices.

## Figures and Tables

**Figure 1 micromachines-12-00588-f001:**
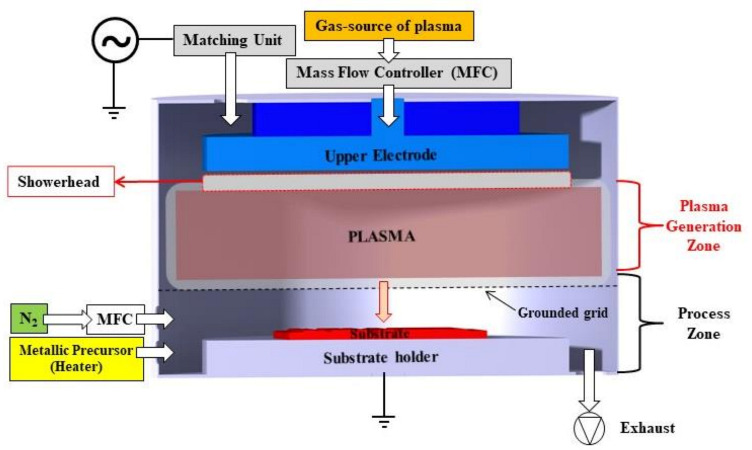
Schematic representation of the capacitively coupled plasma reactor used for plasma-enhanced atomic layer deposition (ALD) processes used to grown TiO_2_/Al_2_O_3_ nanolaminates films.

**Figure 2 micromachines-12-00588-f002:**
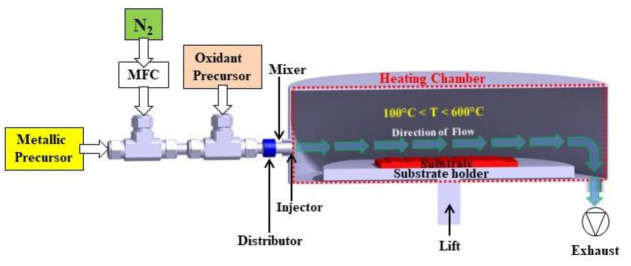
Schematic representation of low-volume crossflow atomic layer deposition (ALD) reactor operating in thermal mode used to grown TiO_2_/Al_2_O_3_ nanolaminates films in our previous work [24].

**Figure 3 micromachines-12-00588-f003:**
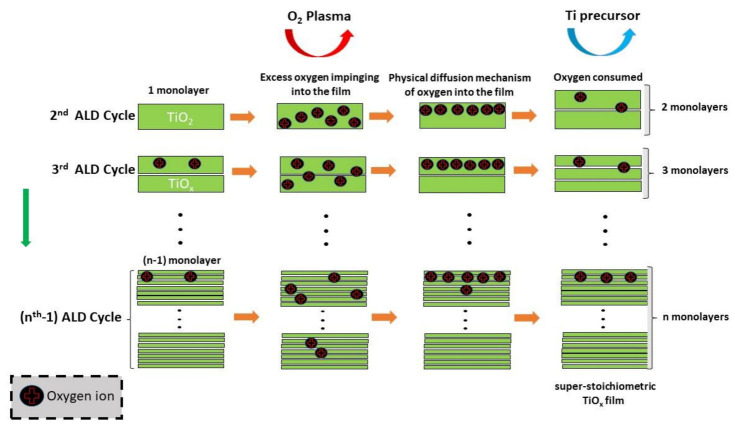
The proposed mechanism of TiO_2_ superstoichiometric film growth during each PEALD cycle for both TTIP and TMA precursors.

**Figure 4 micromachines-12-00588-f004:**
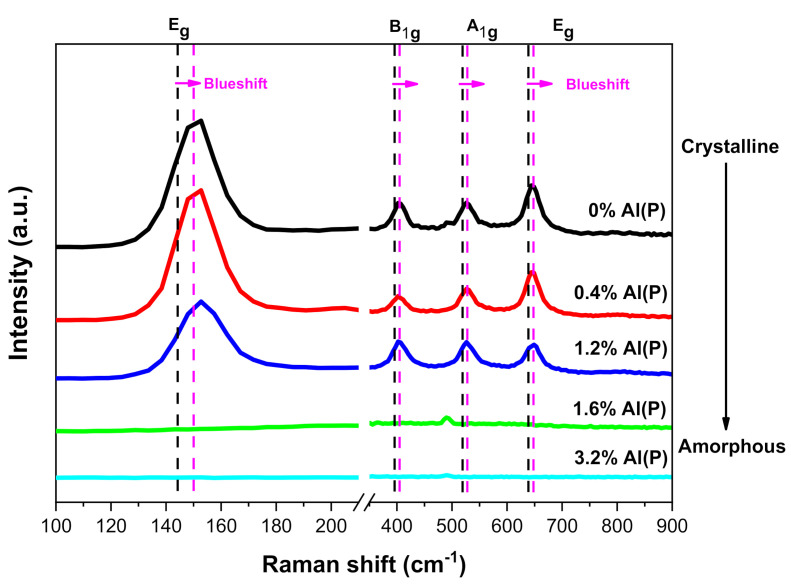
Raman spectra for TiO_2_ film and TiO_2_/Al_2_O_3_ nanolaminates deposited under the pulse ratio from 0.000 to 0.032 grown on p-type <100> Si.

**Figure 5 micromachines-12-00588-f005:**
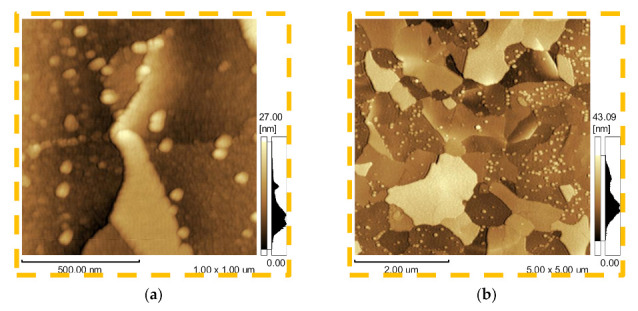
AFM images of the TiO_2_/Al_2_O_3_ nanolaminates for pulse ratio from 0.004 to 0.032 with magnification of 1 × 1 μm^2^ (Figures (**a**,**c**,**e**,**g**)) and 5 × 5 μm^2^ (Figures (**b**,**d**,**f**,**h**)). (**a**,**b**) 0.4% Al(P) samples with pulse ratio and supercycle of 0.004 and 270/1, respectively; (**c**,**d**) 1.2% Al(P) samples with 0.012 and 90/1; (**e**,**f**) 1.6% Al(P) samples with 0.016 and 60/1; and (**g**,**h**) 3.2% Al(P) samples with 0.032 and 30/1.

**Figure 6 micromachines-12-00588-f006:**
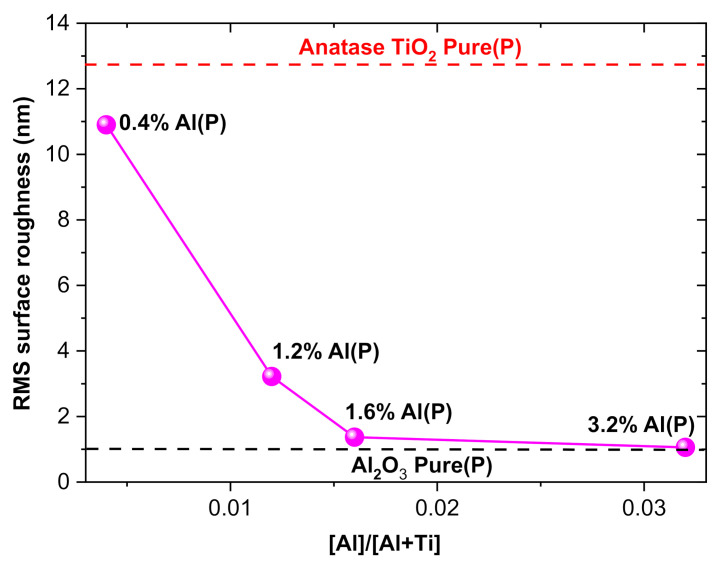
RMS surface roughness of the TiO_2_/Al_2_O_3_ nanolaminates grown on p-type <100> Si under pulse ratio ranging from 0.004 to 0.032.

**Figure 7 micromachines-12-00588-f007:**
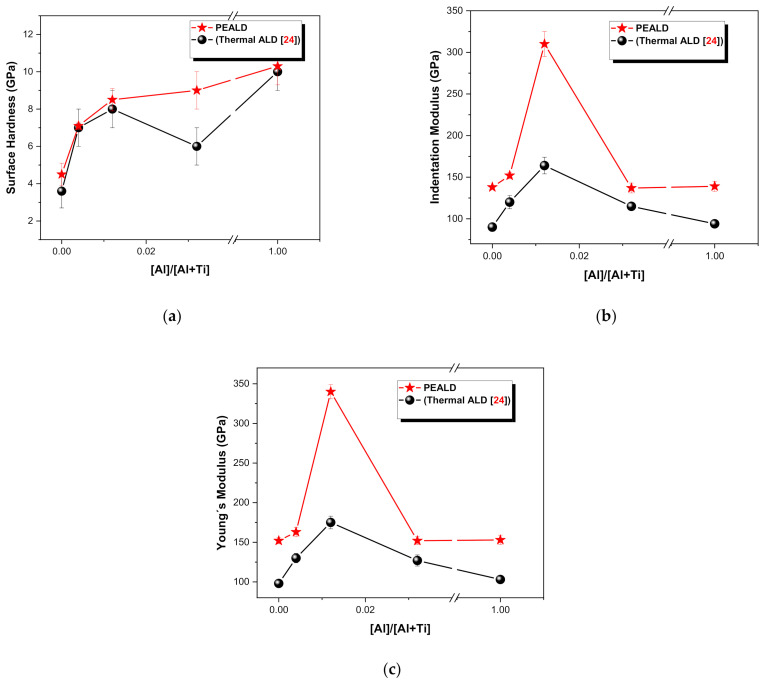
Mechanical properties of the TiO_2_, Al_2_O_3_, and TiO_2_/Al_2_O_3_ nanolaminate thin films measured by nanoindentation technique as a function of pulse ratio ([Al]/[Al + Ti]). (**a**) Surface hardness; (**b**) Indentation modulus; (**c**) Young´s modulus.

**Figure 8 micromachines-12-00588-f008:**
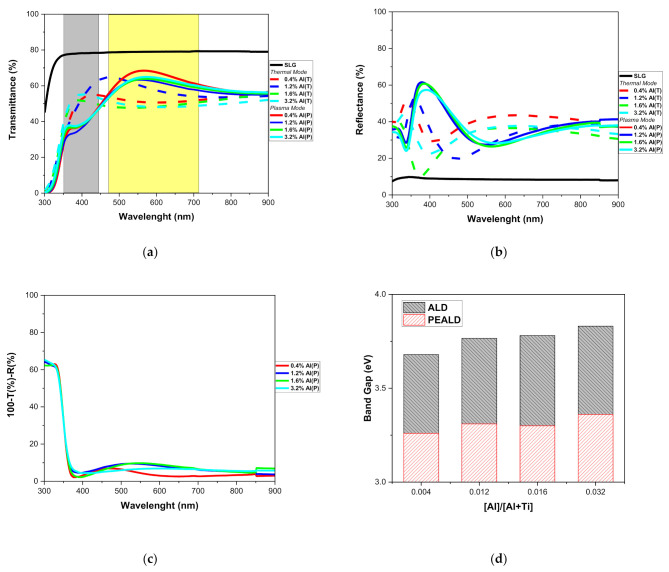
Optical properties: (**a**) Transmittance spectra; (**b**) Reflectance spectra; (**c**) Optical losses spectra; (**d**) Band gap as function of pulse ratio.

**Table 1 micromachines-12-00588-t001:** Process parameters for the growth of TiO_2_, Al_2_O_3_, and TiO_2_/Al_2_O_3_ nanolaminates. All thin films were grown at 250 °C for thermal ALD and PEALD.

ALD ^1^ and PEALD Samples	Pulse Ratio [Al]/[Al + Ti]	Supercycle [ALD TiO_2_ Cycles]/[ALD Al_2_O_3_ Cycles]
0% Al(T) and 0% Al(P)	0	TiO_2_
0.4% Al(T) and 0.4% Al(P)	0.004	270/1
1.2% Al(T) and 1.2% Al(P)	0.012	90/1
1.6% Al(T) and 1.6% Al(P)	0.016	60/1
3.2% Al(T) and 3.2% Al(P)	0.032	30/1
Al_2_O_3_ (T) and Al_2_O_3_ (P)	1	Al_2_O_3_

^1^ Samples related with the previous work [24] grown on thermal mode.

**Table 2 micromachines-12-00588-t002:** Elemental composition, film thickness, and growth per cycle (GPC) of TiO_2_/Al_2_O_3_ nanolaminates grown on p-type <100> Si obtained through RBS spectra by MultiSIMNRA software. The table shows the results of our previous work [24] for comparison.

Sample	Ti (%)	O (%)	Al (%)	Thickness (nm)	GPC (nm/Cycle)
0.4% Al(P)	30 ± 1	70 ± 1	1 ± 1	130 ± 1	0.048 ± 0.005
1.2% Al(P)	31 ± 1	67 ± 1	2 ± 1	130 ± 1	0.048 ± 0.005
1.6% Al(P)	32 ± 1	65 ± 1	3 ± 1	143 ± 1	0.052 ± 0.005
3.2% Al(P)	34 ± 1	60 ± 1	6 ± 1	160 ± 1	0.057 ± 0.005
0.4% Al(T)	37 ± 1	60 ± 1	3 ± 1	88 ± 1	0.032 ± 0.005
1.2% Al(T)	31 ± 1	63 ± 1	6 ± 1	83 ± 1	0.030 ± 0.005
1.6% Al(T)	27 ± 1	64 ± 1	9 ± 1	76 ± 1	0.028 ± 0.005
3.2% Al(T)	25 ± 1	64 ± 1	11 ± 1	83 ± 1	0.030 ± 0.005

**Table 3 micromachines-12-00588-t003:** Peak position, full width at half maximum (FWHM), and integrated area of the Raman-active modes of crystalline samples. It was used the Lorentzian equations to fit the Raman spectra and calculated the peak position, FWHM, and integrated area.

Crystalline Samples	Raman-Active Modes ^1^ (cm^−1^)	Peak Position (cm^−1^)	FWHM (cm^−1^)	Integrated Area (a.u.)
0% Al(P)	E_g_	151 ± 1	19 ± 1	44.200
	B_1g_	404 ± 1	26 ± 1	11.900
	A_1g_	527 ± 1	26 ± 1	10.700
	E_g_	647 ± 1	31 ± 1	23.900
0.4% Al(P)	E_g_	152 ± 1	19 ± 1	45.600
	B_1g_	403 ± 1	26 ± 2	6.800
	A_1g_	527 ± 1	26 ± 1	9.500
	E_g_	647 ± 1	30 ± 1	21.250
1.2% Al(P)	E_g_	153 ± 1	21 ± 1	29.500
	B_1g_	405 ± 1	31 ± 1	14.000
	A_1g_	528 ± 1	29 ± 1	11.700
	E_g_	648 ± 1	30 ± 1	12.700

^1^ Raman-active modes associated to anatase structure: A_1g_ (519 cm^−1^), B_1g_ (397 cm^−1^), and E_g_ (144 and 636 cm^−1^).

**Table 4 micromachines-12-00588-t004:** Summary of the principal works that explore the fundamental characterizations of TiO_2_/Al_2_O_3_ nanolaminates.

Main Results and Comparison with Present Work	(Precursors) (ALD Window) (Substrate Type)	Reference
(i) Amorphous nanolaminates were grown in a bilayer stack (~40 nm of thickness) and a 5 tier multilayer stack (~55 nm of thickness). (ii) The surface hardness of the bilayer was found to be 6 GPa. This result is in agreement with our results that varied from 7 to 9 GPa. (iii) Surface roughness was maintained below 1 nm. These results show that for higher concentrations of Al, the roughness is lower, which is in agreement with our results.	(TiCl_4_/TMA/H_2_O) (100–200 °C) (Silicon; SLG; Polycarbonate)	[86]
(i) It has been shown that nanolaminates are endowed with polycrystalline TiO_2_ properties in the case of TiO_2_ layer thickness more significant than some limit value (20 cycles Al_2_O_3_ and 600 cycles TiO_2_) and become amorphous when the intermediate layers of Al_2_O_3_ (100 cycles Al_2_O_3_ and 450 cycle TiO_2_) increase its thickness. These results show that for higher concentrations of Al, the nanolaminates’ crystallinity becomes amorphous, following our results. It is worth mentioning that we used a different configuration of doping TiO_2_ with Al and obtained the same behavior.	(TTIP/TMA/H_2_O) (250 °C) (Silicon; ITO on Glass)	[87]
(i) Leakage currents for nanolaminates and mixtures have the lowest leakage for all equivalent oxide thickness values. (ii) Currents in the films became strongly affected by chemical and structural defects induced by the deposition process of Al-doped.	(TiCl_4_/TMA/H_2_O) (300–400 °C) (n-Si (100) precovered with 0.6 nm thick SiN_x_; p-Si (100) with 1.1 nm thick SiO_2_)	[88]
(i) The dielectric constants of the Al-doped TiO_2_ films are lower than that of the un-doped TiO_2_ films and decreased with the increase of Al concentration. (ii) Current density of Al-doped TiO_2_ films increased at high applied voltage when the Al concentration in the films was lower.	(TTIP/TMA/O_3_) (200–230 °C) (Silicon; Sputtered Ru and Pt)	[89]
(i) The adsorption of the Ti precursor on the growth surface became less active after the incorporation of Al. This behavior decreased the growth rate of TiO_2_ films doped with Al. It is noteworthy that this behavior is opposite to that achieved in our work, probably due to the O_2_ plasma used as an oxidizing precursor that activates a more significant number of sites on the surface and increases the growth rate of nanolaminates.	(TTIP/TMA/O_3_) (250 °C) (Ru(30 nm)/Ta_2_O_5_(8 nm)/SiO_2_(100 nm)/Si)	[90,91]
(i) The refractive index decreases with the increase of Al_2_O_3_ within the TiO_2_/Al_2_O_3_ nanolaminates, which is in line with our results.	(TiCl_4_/TMA/O_3_) (250 °C) (Corning glass slides and silicon (100) pieces with thin native oxide)	[92]
(i) It was showed that in the supercycle 60/1, the nanolaminates grown on Si are crystallines in the anatase phase and become amorphous at the supercycle 15/1. This result is in line with our work	(TiCl_4_/TMA/O_3_) (350 °C) (Si and RuO_2_)	[93]
(i) nanolaminates are essentially composed by amorphous Al_2_O_3_ and small TiO_2_ crystalline regions for 2, 5, 10, and 20 bilayers composition; (ii) the average transmittance is between 70–95%. It was observed a shift in the maximum transmittance on a range of 375–450 nm wavelengths. In our work, it shifted to 460–750 nm wavelengths with the average transmittance between 60–70%; (iii) the band gap had results similar to our work; (iv) The surface hardness maintained approximately 9 GPa for 2, 5, 10, and 20 bilayers composition. This result is in line with our results; (v) the Young´s and indentation modulus maintained approximately 150 GPa for 2, 5, 10, and 20 bilayers composition. This result is in line with our results;	(TiCl_4_/TMA/H_2_O) (200 °C) (p-doped Si (100) and glass substrates)	[94]
(i) Four sets of different samples were manufactured, one parameter is varied at a time: (a) the growth temperature, (b) the titanium dioxide fraction from 0% to 100%, (c) the bilayer thickness of 0.1 at 100 nm, and (d) the thickness of the nanolaminate from 20 to 300 nm. In all cases the surface hardness, Young´s and indentation modulus maintained approximately 8, 150, and 150 GPa, respectively.	(TiCl_4_/TMA/H_2_O) (110–300 °C) (p-type (100) silicon wafers)	[49]
(i) It was showed that the GPC of Al doped TiO_2_ films increased by ~10% compared to the growth of pure TiO_2_ film by the O_2_ plasma process. This is in line with our results; however, it is in contrast to the lower GPC shown in previous works by the same authors when using the O_3_-based process [91,92].	(TTIP/TMA/O_2_ or N_2_O plasma) (250 °C) (Ru(30 nm)/Ta_2_O_5_(8 nm)/SiO_2_(100 nm)/Si)	[60]

## Data Availability

The data that support the findings of this study are available from the corresponding author upon reasonable request.

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
