# Peer review of "Effect of Plasma-Enhanced Atomic Layer Deposition on Oxygen Overabundance and Its Influence on the Morphological, Optical, Structural, and Mechanical Properties of Al-Doped TiO2 Coating"

_micromachines, 2021, doi:10.3390/mi12060588_

Round 1
Reviewer 1 Report (Previous Reviewer 1)
The reviewer thinks the response of a reviewer's opinion has been properly answered and addressed in the revised manuscript. Thus, the reviewer recommends the publication after English proofreading.
Reviewer 2 Report (Previous Reviewer 3)
The authors carried out the necessary changes and i believe the paper can be accepted in its present form.
(Previous Reviewer 2 didn't has time to review it again, and the Academic Editor helped to check the response and revised manuscript.)
This manuscript is a resubmission of an earlier submission. The following is a list of the peer review reports and author responses from that submission.
Round 1
Reviewer 1 Report
The authors have investigated the chemical, structural, morphological, and optical properties of TiO2/Al2O3 PE-ALD-grown on Si and glass. They modulated the ratio of PEALD TiO2 and Al2O3. The reviewer thinks it is not suitable for the publication of this manuscript to Micro Machines.
- The reviewer can hardly find what is the novelty of this work. First of all, a study on TiO2/Al2O3 nanolaminates can be found in a bunch of reports, probably more than 100 papers (https://doi.org/10.1016/j.tsf.2012.08.023, https://doi.org/10.1021/am500458d, https://doi.org/10.1021/acs.jpcc.5b06745, https://doi.org/10.1021/acsami.5b00677, https://doi.org/10.1016/j.tsf.2017.02.050, https://doi.org/10.1016/j.jcrysgro.2004.10.007, https://doi.org/10.1021/acs.langmuir.9b03988, etc.) However, at the Introduction part, they did not mention why important this work is. Moreover, I was surprised that they even did not mention why they choose a material system, Al2O3/TiO2, for sensor application. They mentioned the growth mechanism of Al2O3/TiO2 is needed to study, but there have been studies on that.
- They specifically named the sample, but it is hard to figure out what they are. The reviewer recommends not to use specific names but to use a common form.
- In Table 1, “pure Al2O3” is wrong. “Pure” means that it does not have any impurity. It should be changed to “single Al2O3”.
- The critical thing is that the reviewer is not convinced of the proposed mechanism of growth shown in Figure 3. To verify their proposed mechanism, they should need to support that with experimental results (not just with their explanation). Especially, oxygen impinging during PE-ALD has not been well known. To the reviewer, the mechanism sounds just a hypothesis. The reviewer is wondering if they observed oxygen stored near the surface during ALD process. Also, this proposed mechanism should be correlated with other film properties they measured.
- In Figure 5, why do AFM images (e)-(h) show a stripe pattern? Are they from the film itself or from measurement condition?
- In Figure 7, they did not mention why different behavior was observed depending on the ratio of films. For example, in Figure 7(a), why the sudden drop is observed, and in Figure 7(b) and 7(c) why the sudden increase? They need to add discussions on that. Better properties for PE-ALD one compared to thermal ALD could be attributed to the super-stoichiometric film, they mentioned. However, the reviewers cannot find any correlation between super-stoichiometry and mechanical properties.
- The authors mentioned this work is for the application of devices and sensing. However, the reviewer cannot find why this work including AFM, RBS, RAMAN, Optical properties.
Response to Reviewer 1
The authors have investigated the chemical, structural, morphological, and optical properties of TiO2/Al2O3 PE-ALD-grown on Si and glass. They modulated the ratio of PEALD TiO2 and Al2O3. The reviewer thinks it is not suitable for the publication of this manuscript to Micro Machines.
Comment 1: The reviewer can hardly find what is the novelty of this work. First of all, a study on TiO2/Al2O3 nanolaminates can be found in a bunch of reports, probably more than 100 papers (https://doi.org/10.1016/j.tsf.2012.08.023, https://doi.org/10.1021/am500458d, https://doi.org/10.1021/acs.jpcc.5b06745, https://doi.org/10.1021/acsami.5b00677, https://doi.org/10.1016/j.tsf.2017.02.050, https://doi.org/10.1016/j.jcrysgro.2004.10.007, https://doi.org/10.1021/acs.langmuir.9b03988, etc.) However, at the Introduction part, they did not mention why important this work is. Moreover, I was surprised that they even did not mention why they choose a material system, Al2O3/TiO2, for sensor application. They mentioned the growth mechanism of Al2O3/TiO2 is needed to study, but there have been studies on that.
Response: Thank you for pointing this out. We added the subsection “3.4. Results comparison with previous studies from other research teams” with a Table to a better understanding and clarify the work. We added in line 357 the following sentence “This section summarizes the main works related to fundamental studies with TiO2 doped with Al and stacks of TiO2/Al2O3 films, both called nanolaminates. The real challenge was to find works from other teams that could be compared to our current work. Our first challenge is related to the fact that each type of TiO2/Al2O3 stacking and all types of Al doping form unique films with specific properties, structures and growth forms. Another difficulty faced is due to the large number of articles that study the application of these materials (more than 100 works were found), ranging from microelectronics [54,61,62,84] to application in tunable color coating [38], which makes it difficult to compare with our work . Finally, we found about 11 works that can be compared with our work. These works vary between the years of 2004 and 2021 and are shown in Table 4 [86-95]. Among these works, only one work that used O2 plasma can be compared with our characterizations [60], that is, there is a lack of works with TiO2/Al2O3 nanolaminates that study fundamental properties and that use O2 plasma as a ligand precursor. Therefore, further studies of PEALD are necessary for the growth of nanolaminates.”
Comment 2: They specifically named the sample, but it is hard to figure out what they are. The reviewer recommends not to use specific names but to use a common form.
Response: Thank you for pointing this out. It was added the acronyms based in concentration of Al in the thin films. We added in line 129 the following sentence “Table 1 summarizes the supercycle and the corresponding pulse ratio utilized in this work. TiO2/Al2O3 nanolaminate films were grown under the following conditions of pulse ratio ([Al]/[Al+Ti]): 0 (sample 0% Al(P) (pure TiO2)); 0,004 (sample 0,4% Al(P)); 0,012 (sample 1,2% Al(P)); 0,016 (sample 1,6% Al(P)); 0,032 (sample 3,2% Al(P)); and 1 (sample Al2O3 pure(P)). From our previous work using thermal ALD [24], the samples that were grown using the same pulse ratio condition were labeled as 0% Al(T); 0,4% Al(T); 1,2% Al(T); sample 1,6% Al(T); 3,2% Al(T); and Al2O3 pure(T), respectively. The (P) represents samples grown on plasma mode, and (T) represents samples grown on thermal mode.”
Comment 3: In Table 1, “pure Al2O3” is wrong. “Pure” means that it does not have any impurity. It should be changed to “single Al2O3”.
Response: Thank you so much for catching this error; we extracted the word “pure” along overall the manuscript.
Comment 4: The critical thing is that the reviewer is not convinced of the proposed mechanism of growth shown in Figure 3. To verify their proposed mechanism, they should need to support that with experimental results (not just with their explanation). Especially, oxygen impinging during PEALD has not been well known. To the reviewer, the mechanism sounds just a hypothesis. The reviewer is wondering if they observed oxygen stored near the surface during ALD process. Also, this proposed mechanism should be correlated with other film properties they measured.
Response: Thank you for pointing this out. We believe that this mechanism is not speculative. Previously, Professor Stacey Bent's group showed similar behavior and mechanism for Fe2O3 Atomic Layer Deposition using tert-Butylferrocene and O3(https://onlinelibrary.wiley.com/doi/abs/10.1002/admi.202000318), and our group showed a similar mechanism for TiO2 and O2 Plasma (https://www.frontiersin.org/articles/10.3389/ fmech.2020.551085 / full).
Comment 5: In Figure 5, why do AFM images (e)-(h) show a stripe pattern? Are they from the film itself or from measurement condition?
Response: Thank you for pointing this out. These patterns are from the films.
Comment 6: In Figure 7, they did not mention why different behavior was observed depending on the ratio of films. For example, in Figure 7(a), why the sudden drop is observed, and in Figure 7(b) and 7(c) why the sudden increase? They need to add discussions on that. Better properties for PE-ALD one compared to thermal ALD could be attributed to the superstoichiometric film, they mentioned. However, the reviewers cannot find any correlation between superstoichiometry and mechanical properties.
Response: Thank you for pointing this out. We added in line 312 the following sentence “Therefore, it can be suggested that the surface hardness (Figure 7a) is dependent on the increase of the amount of Al in the nanolaminates. On the other hand, it can be suggested that the indentation and Young´s modulus are dependent on the superstoichiometry, as can be seen in 0,4% Al(P) and 1,2% Al(P) samples (Figure 7b and 7c), where occurs a sudden increase” to a better explanation of the graphs.
Comment 7: The authors mentioned this work is for the application of devices and sensing. However, the reviewer cannot find why this work including AFM, RBS, RAMAN, Optical properties.
Response: Thank you for pointing this out. The Title, Abstract, and Introduction were rewritten to solve this issue.
Reviewer 2 Report
This is an interesting article entitled, “Material Properties of PEALD TiO2/Al2O3 Nanolaminates for Device and Sensing Applications”, where the authors have discussed their experimental approach to optimize properties of TiO2/Al2O3 nanolaminates. I have few concerns about the overall approach for the work as mentioned below.
- Abstract seems to be vaguely written and can be improved for clarification. This must be a standalone content by itself.
- The title mentions the use of nanolaminates in sensing applications, but this is not supported by any data in the manuscript.
- The authors have shown a speculative mechanism for film growth, but they need to support this by conducting detailed surface/cross- sectional morphological and chemical characterization.
- How is the interface between TiO2 and Al2O3? How does it change with growth?
Response to Reviewer 2
This is an interesting article entitled, “Material Properties of PEALD TiO2/Al2O3 Nanolaminates for Device and Sensing Applications”, where the authors have discussed their experimental approach to optimize properties of TiO2/Al2O3 nanolaminates. I have few concerns about the overall approach for the work as mentioned below. Comment 1: Abstract seems to be vaguely written and can be improved for clarification. This must be a standalone content by itself. Response: Thank you for pointing this out. The “Abstract” was rewritten.
Comment 2: The title mentions the use of nanolaminates in sensing applications, but this is not supported by any data in the manuscript.
Response: Thank you for pointing this out. The “title” was rewritten to better adequate to Introduction.
Comment 3: The authors have shown a speculative mechanism for film growth, but they need to support this by conducting detailed surface/cross- sectional morphological and chemical characterization.
Response: Thank you for pointing this out. We believe that this mechanism is not speculative. Previously, Professor Stacey Bent's group showed similar behavior and mechanism for Fe2O3 Atomic Layer Deposition using tert-Butylferrocene and O3 (https://onlinelibrary.wiley.com/doi/abs/10.1002/admi.202000318), and our group showed a similar mechanism for TiO2 and O2 Plasma (https://www.frontiersin.org/articles/10.3389/ fmech.2020.551085 / full).
Comment 4: How is the interface between TiO2 and Al2O3? How does it change with growth?
Response: Thank you for pointing this out. These responses were shown in our previous work (https://iopscience.iop.org/article/10.1088/0022-3727/49/37/375301).
Reviewer 3 Report
The manuscript provides interesting data about the chemical, structural, morphological, mechanical and optical properties of TiO2/Al2O3 nanolaminates grown by plasma-enhanced atomic layer deposition (PEALD). Overall, the paper is well written and properly presented. Many studies already described the materials presented in this work, but the latter gives new insights about the properties of these specific nanolaminates/doped TiO2 films. Although the work is interesting – some changes should be carried out and properly addressed prior to publication.
- The text is easy is to read throughout the paper. However, the description of the samples PR4, 12, 16, 32, and R4,12,16,32 should be more understandable. I would suggest to use different acronyms to facilitate the reading.
- A major choice of wording appears very inappropriate. The authors use the term “controlled poisoning effect” instead of the usually used term “doping” in the ALD community. This is wrong and very confusing. In general, the poisoning effect is linked to metalorganic ligands of the precursors sticking at the substrate surface, hence “poisoning” it. In this work, the addition of TMA steps in the supercycles of the process to synthesize TiO2/Al2O3 layers is not a poisoning, it is a so-called doping. The author should replace the term “controlled poisoning” by “doping” for the layers prepared, except for the pure TiO2 and Al2O3 (obviously). This should be done overall the manuscript.
- The Micromachines journal is aimed to report on the science and technology of small structures, devices and systems. The author should add a paragraph or rephrase the introduction and the conclusion in order to show a direct link between the data they report and the focus of the journal. For example, they should describe how the optical/roughness/mechanical properties of the films reported are important for certain devices.
- In the introduction, the author indicates that ALD is allowing applications in a broad range of fields, such as micro and nanoelectronics, biomedical engineering, on food packaging against corrosion, fuel cells, solar cells, anti-tarnish coatings on jewels surfaces, and smart textiles. Two key novel applications are missing and should be added and referred to: Membranes and optoelectronics (Weber et al., Journal of Applied Physics 126 (4), 041101 (2019), Chemistry of Materials 30 (21), 7368-7390 (2018) / Mattinnen Chem. Mater. , 31, 15, 5713–5724 (2019)).
- In page 2, the authors claim that studies are needed to understand essential aspects of the chemical, morphological, mechanical and optical properties of the Al2O3/TiO2 thin films prepared. However, it is important to mention that TiO2, Al2O3, as well as doped Al doped TiO2 have been extensively studied in the past. These studies should be referreed to. The authors could see for example the well cited review of Puurunen for Al2O3 (Journal of applied physics, 2005), and the work of Choi on plasma-enhanced ALD of TiO2 and Al-Doped TiO2 Films Using O2 Reactants (Choi et al 2009 J. Electrochem. Soc. 156 G138), among others. I would recommend the authors to compare their present work with previous studies from other research teams (not only to their own work). The data compared could for example be included in an additional Figure/ Table.
- The fact that the films were grown at 250C should be indicated in the main text of the Materials and Methods section, not only in the Table 1 caption.
Response to Reviewer 3
Comment 1: The text is easy is to read throughout the paper. However, the description of the samples PR4, 12, 16, 32, and R4,12,16,32 should be more understandable. I would suggest to use different acronyms to facilitate the reading.
Response: Thank you for pointing this out. It was added the acronyms based in concentration of Al in the thin films. We added in line 129 the following sentence “Table 1 summarizes the supercycle and the corresponding pulse ratio utilized in this work. TiO2/Al2O3 nanolaminate films were grown under the following conditions of pulse ratio ([Al]/[Al+Ti]): 0 (sample 0% Al(P) (pure TiO2)); 0,004 (sample 0,4% Al(P)); 0,012 (sample 1,2% Al(P)); 0,016 (sample 1,6% Al(P)); 0,032 (sample 3,2% Al(P)); and 1 (sample Al2O3 pure(P)). From our previous work using thermal ALD [24], the samples that were grown using the same pulse ratio condition were labeled as 0% Al(T); 0,4% Al(T); 1,2% Al(T); sample 1,6% Al(T); 3,2% Al(T); and Al2O3 pure(T), respectively. The (P) represents samples grown on plasma mode, and (T) represents samples grown on thermal mode.”
Comment 2: A major choice of wording appears very inappropriate. The authors use the term “controlled poisoning effect” instead of the usually used term “doping” in the ALD community. This is wrong and very confusing. In general, the poisoning effect is linked to metalorganic ligands of the precursors sticking at the substrate surface, hence “poisoning” it. In this work, the addition of TMA steps in the supercycles of the process to synthesize TiO2/Al2O3 layers is not a poisoning, it is a so-called doping. The author should replace the term “controlled poisoning” by “doping” for the layers prepared, except for the pure TiO2 and Al2O3 (obviously). This should be done overall the manuscript.
Response: Thank you so much for catching this error; we changed the word “poisoning” for “doping” along overall the manuscript.
Comment 3: The Micromachines journal is aimed to report on the science and technology of small structures, devices and systems. The author should add a paragraph or rephrase the introduction and the conclusion in order to show a direct link between the data they report and the focus of the journal. For example, they should describe how the optical/roughness/mechanical properties of the films reported are important for certain devices.
Response: Thank you for pointing this out. We added in line 69 (Introduction) the following sentence “For example, roughness, which is a morphological property, can be used to control the mobility of pair hole-electron in metal-insulator-semiconductor (MIS) devices [42], which is essential control to enhance the power-conversion efficiency (PCE) on MIS solar cells devices [43]. A Mechanical property considered fundamental in all devices and sensors is the protection against corrosion, which increases the device's lifetime [44]. On the other hand, sensors based on optical parameters can be fabricated through control of refractive index, which is an optical property that can be used to produce a high-quality resonant waveguide grating (RWG) that is used to fabricate fluorescence biosensors, and photodetectors, beyond other devices [45]. Therefore, it is essential to understand the fundamental properties of thin films.” We added in line 375 (Conclusion) the following sentence “The morphological modification is essential to control the mobility of pair hole-electron in metal-insulator-semiconductor devices, being crucial to enhance the power-conversion efficiency in MIS solar cells. The property called protection against corrosion is essential to increases the device's lifetime. The optical property improvement can be used to produce a high-quality resonant waveguide grating used to fabricate fluorescence biosensors and photodetectors beyond other devices.”
Comment 4: In the introduction, the author indicates that ALD is allowing applications in a broad range of fields, such as micro and nanoelectronics, biomedical engineering, on food packaging against corrosion, fuel cells, solar cells, anti-tarnish coatings on jewels surfaces, and smart textiles. Two key novel applications are missing and should be added and referred to: Membranes and optoelectronics (Weber et al., Journal of Applied Physics 126 (4), 041101 (2019), Chemistry of Materials 30 (21), 7368- 7390 (2018) / Mattinnen Chem. Mater. , 31, 15, 5713–5724 (2019)).
Response: Thank you for pointing this out. The two novel applications for the ALD technique and the suggested references have been added in line 65, we add the suggested references numbered as [39-41].
Comment 5: In page 2, the authors claim that studies are needed to understand essential aspects of the chemical, morphological, mechanical and optical properties of the Al2O3/TiO2 thin films prepared. However, it is important to mention that TiO2, Al2O3, as well as doped Al doped TiO2 have been extensively studied in the past. These studies should be referreed to. The authors could see for example the well cited review of Puurunen for Al2O3 (Journal of applied physics, 2005), and the work of Choi on plasma-enhanced ALD of TiO2 and Al-Doped TiO2 Films Using O2 Reactants (Choi et al 2009 J. Electrochem. Soc. 156 G138), among others. I would recommend the authors to compare their present work with previous studies from other research teams (not only to their own work). The data compared could for example be included in an additional Figure/ Table.
Response: Thank you for pointing this out. We rewrote the all "Introduction", and We added the references suggested.
Comment 6: The fact that the films were grown at 250ºC should be indicated in the main text of the Materials and Methods section, not only in the Table 1 caption.
Response: Thank you for pointing this out. We added the process temperature of 250ºC in line 116.